# A Study on the Mechanism and Kinetics of Ultrasound-Enhanced Sulfuric Acid Leaching for Zinc Extraction from Zinc Oxide Dust

**DOI:** 10.3390/ma15175969

**Published:** 2022-08-29

**Authors:** Xuemei Zheng, Shiwei Li, Bingguo Liu, Libo Zhang, Aiyuan Ma

**Affiliations:** 1Faculty of Metallurgical and Energy Engineering, Kunming University of Science and Technology, Kunming 650093, China; 2School of Chemistry and Materials Engineering, Liupanshui Normal University, Liupanshui 553004, China; 3Key Laboratory of Unconventional Metallurgy, Kunming University of Science and Technology, Kuming 650093, China

**Keywords:** zinc oxide dust (ZOD), ultrasound-enhanced leaching, zinc extraction

## Abstract

As an important secondary zinc resource, large-scale reserves of zinc oxide dust (ZOD) from a wide range of sources is of high comprehensive recycling value. Therefore, an experimental study on ultrasound-enhanced sulfuric acid leaching for zinc extraction from zinc oxide dust was carried out to investigate the effects of various factors such as ultrasonic power, reaction time, sulfuric acid concentration, and liquid–solid ratio on zinc leaching rate. The results show that the zinc leaching rate under ultrasound reached 91.16% at a temperature of 25 °C, ultrasonic power 500 W, sulfuric acid concentration 140 g/L, liquid–solid ratio 5:1, rotating speed 100 r/min, and leaching time 30 min. Compared with the conventional leaching method (leaching rate: 85.36%), the method under ultrasound increased the zinc leaching rate by 5.8%. In a kinetic analysis of the ultrasound-enhanced sulfuric acid leaching of zinc oxide dust, the initial apparent activation energy of the reaction was 6.90 kJ/mol, indicating that the ultrasound-enhanced leaching process was controlled by the mixed solid product layers. Furthermore, the leached residue was characterized by XRD and SEM-EDS, and the results show that, with ultrasonic waves, the encapsulated mineral particles were dissociated, and the dissolution of ZnO was enhanced. Mostly, the zinc in leached residue existed in the forms of ZnFe_2_O_4_, Zn_2_SiO_4_, and ZnS.

## 1. Introduction

As an extensive metallic element in the world, zinc is broadly used in various fields such as the automobile, construction, shipbuilding, and aerospace industries as plated zinc, zinc-based alloy, and zinc oxide, etc. Mostly, zinc ores are composed of zinc sulfide, zinc oxide, or a mixture thereof. In recent years, zinc resources in China have generally been characterized by less rich ores and more low-grade ores, fewer large mines, and more small or medium mines which are difficult to exploit. At present, 70% of zinc in the world comes from zinc ore resources, while the remaining 30% comes from secondary zinc resources [1,2,3,4].

There are various sources of secondary zinc resources, e.g., hot galvanizing slag [5] and zinc ash [6], smelting slag (mud) and zinc-containing dust produced by the copper, lead, and zinc smelting industry [7,8,9], electric arc furnace (EAF) dust (mud) from the iron and steel industry [10,11], zinc-loaded waste catalysts [12,13,14], waste zinc manganese batteries [15], and circuit boards [16]. To date, more than 2 million tons of zinc have been recovered from secondary zinc resources, and at the same time, the growth rate for the recycling of zinc metals, alloys, and zinc compounds has been three times higher than that for the production of original zinc, indicating that the recovery of secondary zinc resources plays an important role in the recycling economy at present.

Secondary zinc resources are from a wide range of sources [17,18,19,20,21]. For zinc oxide dust from lead and zinc smelting, which is an important secondary zinc resource, the composition is complicated with many coexisting valuable metals and elementary impurities due to the complex zinc hydrometallurgy process for leached residue. In addition to the dissolution and leaching of ZnO in roasted ore as well as the hydrolysis and purification of Fe^3+^ in leach solution for iron removal, with the addition of neutralizing agents such as zinc calcine and lime milk, the changes in pH value and concentrations of some metal ions (e.g., Cu^2+^ > 800 mg/L) may be accompanied by the hydrolysis of copper, cadmium, cobalt, and silicon. The complexity in the composition of slag from the zinc hydrometallurgy process is related to the content of valuable metals and impurities in raw materials (e.g., Zn, Pb, Fe, Cu, Cd, In, Co, Si, As, F, and Cl) and the technical control for the process, and thus, there are often a lot of valuable metals and precious metals in neutral leached residue [22,23,24,25,26].

Mostly, zinc in the leached residue occurs in the forms of ZnO, ZnS, ZnFe_2_O_4_, and ZnSiO_4_, and for the treatment of zinc-bearing leached residue, usually, a high-temperature reduction volatilization method is used to recover the valuable metals in neutral leached residue in the conventional zinc hydrometallurgy process (e.g., blast furnace smelting and smoke furnace smelting, sulfation roasting, chlorination, sulfation roasting, and rotary kiln roasting). The zinc content in zinc oxide dust produced by high-temperature reduction volatilization may reach 60%, with ZnO as the main zinc phase. However, the high-temperature volatilization process is accompanied by many reactions, in which the polymers tend partially to encase the zinc oxide phase, and thus, the zinc leaching rate for zinc oxide dust leaching by conventional acid method [27,28,29], alkali method [30,31], or ammonia method [32,33,34,35] is low. According to Oustadakis et al. [27], the recovery of Zn from EAFD can reach 80% with diluted sulfuric acid leaching. In this case, iron is partially transferred into the solution, and iron leaching reaches 45%. Sethurajan et al. [28] have examined the sulfuric acid leaching of the three different zinc plant leach residues (ZLR). The results showed that a higher temperature and acid concentration are required to leach the maximum Zn from the ZLRs for sulfates, oxides, and ferrite minerals. Fattahi et al. [29] examined the reductive leaching of zinc, cobalt, and manganese from zinc plant residue with dilute sulfuric acid and citric acid. The maximum Co, Mn, and Zn recoveries were 96.43%, 90.26%, and 64.12%, respectively. According to Ashtari et al. [30], 82.4% of the zinc was recovered from the zinc plant residue by the conventional alkaline leaching under NaOH concentration of 9 M, the temperature of 25 °C, time of 45 min, speed of 400 rpm, and S/L ratio of 1/10. They also found that the unreacted core of ZnO particles can be significantly improved by mechanochemical alkali leaching. Zhang et al. [31] proposed a process of primary normal pressure leaching and secondary alkaline pressure leaching zinc from EAF dust, 66.4%, and 88.7% Zn can be leached, respectively. The optimum conditions were a temperature of 70 °C, NaOH concentration of 6 mol/L, L/S ratio of 20 mL/g, and a reaction time of 2 h. Ma et al. [32] recovered zinc from blast furnace dust in the ammonia leaching system containing different leaching agents: ammonium sulphate, ammonium carbonate, ammonium citrate, and ammonia, from which 75.32%, 72.52%, 65.99%, and 31.92% Zn was recovered, respectively. The study found that Zn_2_SiO_4_, ZnS, and ZnFe_2_O_4_ did not leach into the ammonia system, which was one of the main causes of the lower zinc extraction rate [33,34]. In addition, the leaching rate of zinc can be effectively improved by microwave calcification pretreatment [9,35]. 

However, as a new unconventional metallurgical technology [36], ultrasonic metallurgy has been widely used in the comprehensive recycling of valuable metals by many researchers. The ultrasound-enhanced leaching process is mainly reflected in the cavitation effect, mechanical effect, and thermal effect. Under the action of ultrasonic cavitation, cavitation bubbles grow and rupture at a certain sound pressure, forming a local high temperature and high pressure forward flow zone in a tiny space, which promotes the leaching reaction. Under the mechanical action, through agitation and flow in the leaching solution, the ultrasound stirs the liquid intensively, reduces the diffusion resistance, and accelerates the mass and heat transfer to accelerate the diffusion and dissolution process of the medium, Through the thermal effect, the ultrasonic energy is continuously absorbed by the medium and converted into heat energy, which further promotes the reaction. Ultrasound-enhanced leaching provides a very special new physical environment for the difficult or impossible reactions to realize under conventional conditions, and thus, to a certain extent, the leaching conditions are improved, the reaction time is shortened, the recovery rate of valuable metals is increased, and an efficient metallurgical process of energy saving and environmental protection is realized [37,38,39,40,41,42,43,44,45]. Wang et al. [37] reported enhanced zinc leaching kinetics from zinc residues augmented with ultrasound. Similarly, Brunelli [38] reported an ultrasound-assisted leaching process for the recovery of zinc from electric arc furnace (EAF) dust. The research results show that ultrasonic-assisted leaching is a suitable technique to improve the dissolution of ZnFe_2_O_4_, which represents the main obstacle during conventional leaching. In addition, based on the advantages of using ultrasound in strengthening the leaching process, ultrasonic technology has been widely used in the treatment of metals such as copper (Cu) [39], uranium (U) [40], gold (Au) [41], silver (Ag) [42,43], and germanium (Ge) [44,45], and has achieved relatively significant leaching effects.

With zinc oxide dust volatilized from a rotary kiln for lead and zinc smelting as the object, this study introduced an ultrasound-enhanced method with sulfuric acid as the leaching agent to investigate the effects of ultrasonic power (UP), sulfuric acid concentration (C), leaching time (t), liquid–solid ratio (L/S), rotating speed (r), and temperature (T) on the zinc leaching rate of zinc oxide dust, and explore the kinetics of ultrasound-enhanced leaching of zinc oxide dust. At the same time, the leaching mechanism of zinc oxide dust was analyzed by XRD and SEM-EDS.

## 2. Experimental Materials and Characterization

### 2.1. Analysis on the Composition of Raw Materials

The raw materials used in the experiment were from a zinc hydrometallurgy enterprise in Yunnan, China. The chemical composition of the ZOD sample, as shown in Table 1, is complicated, with a zinc content up to 41.37%, coexisting with valuable metal elements, e.g., Pb, Mn, and Cd, a scattered associated metal content as high as 820.8 g/t, as well as large amounts of S, Cl, Si, and Ca. The zinc oxide dust had a high recycling value.

### 2.2. Analysis on Mineral Phase

To determine the forms of various metal elements and impurity components existing in zinc oxide dust, the samples were characterized by XRD (TTRIII Multifunctional X-ray Diffractometer, Rigaku, Japan) and SEM-EDS (XL30ESEM-TMP scanning electron microscope, Philips, The Netherlands). The results are shown in Figure 1, Figure 2 and Figure 3, respectively.

The XRD patterns show that most zinc exists as ZnO, Zn_2_SiO_4_, ZnS, and ZnFe_2_O_4_, and that most lead exists as PbS and PbSO_4_. In addition, there are large amounts of gangues in zinc oxide dust, especially SiO_2_ and CaSiO_3_.

The SEM pattern indicates that ore particles in zinc oxide dust are compact with lots of amorphous flocculent inclusions embedded among the particles. The point and surface analysis results of SEM-EDS (see Figure 2 and Figure 3) show that in zinc oxide dust, most of the zinc is distributed in tiny floccule particles as floccules coexisting with O, Fe, Si, and Zn and lumps coexisting with O, S, Pb, and Ca. The zinc oxide dust was different from the original ore in mineral morphology, material existence form, and mineral surface property. The granular minerals were fused at a high temperature with zinc easy to be enveloped by other valuable metals and gangue components.

### 2.3. Experimental Methods

A certain amount of zinc oxide dust and a certain concentration of prepared sulfuric acid leaching solvent were added to a 300 mL conical flask in a certain ratio, and then, the conical flask was placed on a thermostatic magnetic agitator with a digital display, and an ultrasonic probe was inserted in the thermostatic water bath at a position equivalent to the level of solution in the conical flask. The experimental apparatus for ultrasonic leaching is presented in Figure 4. The flow diagram of the zinc leaching process is shown in Figure 5. During the ultrasonic leaching process, with the increase in ultrasonic power or the prolongation of the ultrasonic leaching time, the temperature of the water bath will increase to a certain extent. To ensure that the leaching temperature remains constant, the temperature of the water bath was adjusted to 2 °C during the experiment. For the ultrasound system, an ultrasonic probe continuously adjustable within power 0~2000 W and resistant to a certain concentration of acid was adopted to provide an ultrasonic field. The ultrasonic power and stirring speed could be controlled and adjusted as required. Filtered after a certain leaching time, the zinc concentration in the leaching solution was determined by EDTA titration method. In the determination process, there may have been interference from Cu^2+^, Al^3+^, and Fe^2+^ on Zn^2+^. To eliminate the interference, saturated thiourea, ascorbic acid, and potassium fluoride solutions were added before adding the xylenol orange indicator. The zinc extraction rate (*η*_Zn_, %) may be calculated by the following formula:(1)$$\eta_{Zn}=\frac{C_{\mathrm{Zn}}\times V}{m\times w_{Zn}^{*}}$$
where *C*_Zn_—Zn concentration in the leaching solution, g/L; V*—the volume of leaching solution*, *L*; *m—mass of the* sample, *g*; and $w_{Zn}^{*}$—percentage of Zn in the sample. 

## 3. Experimental Results and Relevant Analysis

### 3.1. Experimental Study on Ultrasound-Enhanced Leaching Conditions

#### 3.1.1. Effect of Ultrasonic Power on Zinc Leaching Rate

The effect of ultrasonic power on the zinc leaching rate over time was investigated under a sulfuric acid concentration of 100 g/L, a liquid–solid ratio of 4:1, a rotating speed of 100 rpm, and a temperature of 65 °C, and the results are as shown in Figure 6. It can be seen that the zinc leaching rate gradually increases with time in direct proportion. At 0–20 min, the zinc leaching rate rapidly increased with time, and after 20 min, the zinc leaching efficiency was obviously lowered. In addition, the zinc leaching rate increased with increasing ultrasonic power, and when the ultrasonic power was 100 W, the zinc leaching rate was only 51.62%; when it was more than 300 W, the zinc leaching rate significantly increased, to 62.31%, 64.6%, 67.25%, and 68.94%, respectively, with the ultrasonic power of 300–900 W, within a time of 30 min. These results indicate that the ultrasonic power had a significant effect on the leaching rate of zinc. Considering that, when the ultrasonic power exceeded 500 W, the zinc leaching rate was obviously not further advanced, and in combination with the relevant energy consumption, the ultrasonic power of leaching was controlled at 500 W.

#### 3.1.2. Effect of Sulfuric Acid Concentration on Zinc Leaching Rate

The effect of the sulfuric acid concentration on the leaching rate of zinc was investigated under an ultrasonic power of 500 W, a liquid–solid ratio of 4:1, a rotating speed of 100 rpm, and a temperature of 65 ℃. According to Figure 7, the zinc leaching rate also increased with sulfuric acid concentration. With time, the zinc leaching rate was increasing because the hydrogen ion concentration in the reaction increased with the sulfuric acid concentration. Promoting the contact of sulfuric acid with a granular zinc phase in zinc oxide dust was conducive to the leaching of zinc from the dust, but with time, hydrogen ions in the solution were consumed, and thus, the leaching rate was gradually lowered. Considering the high concentration of sulfuric acid, the dissolved Fe^3+^ ions increased accordingly, making it difficult to perform the subsequent treatment of the leaching solution. Therefore, the optimal concentration of sulfuric acid was determined to be 140 g/L.

#### 3.1.3. Effect of Liquid–Solid Ratio on Zinc Leaching Rate

The effect of the liquid–solid ratio on the leaching rate of zinc from the zinc oxide dust was investigated under an ultrasonic power of 500 W, a sulfuric acid concentration of 140 g/L, a rotating speed of 100 rpm, and a temperature of 65 °C. The results shown in Figure 8 demonstrate that with the increase in the liquid–solid ratio, the zinc leaching rate first gradually increases, and then reaches a plateau. The main reason for this was that with the increase in the liquid–solid ratio, the fluidity of ions in and out of the system increased, and thus, the movement and collision of fine particles in the zinc oxide dust, as well as relevant reactions, were further intensified. When the liquid–solid ratio was increased from 2:1 to 4:1, the zinc leaching rate was promoted significantly, but when the liquid–solid ratio was further increased, the zinc leaching effect was obviously compromised, although the zinc leaching rate was still increased to some extent. When the liquid–solid ratio was 5:1, the 30 min leaching rate of zinc from the zinc oxide dust was 91.16%. Considering the increased liquid–solid ratio would compromise the subsequent purification and bring difficulties to the follow-up recovery process, and the optimal liquid–solid ratio was determined as 5:1.

#### 3.1.4. Effect of Rotating Speed on Zinc Leaching Rate

The effect of the rotating speed on the leaching rate of zinc from the zinc oxide dust was investigated under an ultrasonic power of 500 W, a sulfuric acid concentration of 140 g/L, liquid–solid ratio of 5:1, and a temperature of 65 °C. The results were as shown in Figure 9. It can be seen that, as the rotating speed increases, the leaching rate of zinc from the zinc oxide dust gradually increases with time, and ultimately reaches a plateau. If the stirring speed was low, the zinc oxide dust particles dissolved in sulfuric acid solution were easy to settle, which was not conducive to the leaching reaction, while an excessively high stirring speed would increase the energy consumption and the cost of the leaching process. Therefore, it is advisable to control the stirring speed at 100 rpm.

#### 3.1.5. Effect of Temperature on Zinc Leaching Rate

The effect of temperature on the leaching rate of zinc from the zinc oxide dust was investigated under an ultrasonic power of 500 W, a sulfuric acid concentration of 140 g/L, a liquid–solid ratio of 5:1, and a rotating speed of 100 rpm. The results are as shown in Figure 10. Figure 10a shows that the change in temperature has a significant effect on the zinc leaching rate when the time was 0 to 20 min, and the zinc leaching rate increases with the increase in temperature. At different temperatures, the zinc leaching rate increased rapidly with time at first, and after 20–30 min, gradually reached a plateau. This is because, with the increase in leaching temperature, the leaching reaction rate was advanced accordingly, and at the same time, the viscosity of the solution decreased, which was conducive to the diffusion of a leaching solvent and product, and thus, the zinc leaching rate was significant at the beginning, while at a later stage, with the continuous consumption of a leaching agent, the leaching efficiency was gradually lowered, and a further increase in leaching temperature did not affect the dissolution of zinc oxide dust. Figure 10b shows that the zinc leaching rate of zinc oxide dust was less affected by temperature after the leaching time reaches 30 min, and the zinc leaching rates were 91.16% and 92.44% at 25 °C and 75 °C, respectively. Considering that high temperatures may increase the volatilization of acid, resulting in a high acid consumption and increased economic cost, the leaching temperature was controlled at 25 °C, and the 30 min zinc leaching rate reached 91.16%.

### 3.2. Kinetics of Ultrasound-Enhanced Leaching

The leaching of zinc from zinc oxide dust is a process of liquid–solid reaction, and the leaching reaction process may be controlled by the following steps: (i) the diffusion of a reactant or product for the leaching agent through the liquid boundary layer; (ii) the diffusion of a reactant or product for a leaching agent through the solid product layer; (iii) the chemical reaction of the reactant for a leaching agent with the surface of unreacted nuclear material; and (iv) the mixture of the solid film diffusion and interfacial chemical reaction.

An analysis of the raw material showed that the Zn-bearing dust particles was irregular in morphology, with a relatively complex composition, which included a granular zinc oxide phase and gangue particles, and most of the particles wrapped the Zn-bearing phase. Mostly, the zinc was embedded in the gangue mineral, and in the leaching process, the leaching agent was diffused to the gap or crack of gangue, and reacted with a zinc mineral contained in the zinc oxide dust. With the reaction, the interface for reaction continuously shrank into the center of zinc mineral particles, and the by-products or residual solid layer was thickening to enlarge the path for the diffusion of a reactant or product for leaching agent. In addition, the inert solid residue of the gangue tended to wrap the unreacted shrunk nuclei as a factor controlling the zinc leaching rate of zinc-bearing mineral particles. Therefore, a model of shrinking core was used to explore the kinetic behavior for the leaching of zinc from the zinc-baring metallurgical dust.

Based on the model of a shrinking core, when the solid–liquid phase reaction is controlled by diffusion reaction, the leaching kinetics equation of zinc oxide dust particles may be expressed as follows:*k*_d_·*t* = 1 − 2/3*x* − (1 − *x*) ^2/3^(2)
when the solid–liquid phase reaction is controlled by interfacial chemical reaction, the leaching kinetics equation of zinc oxide dust particles may be expressed as follows:*k*_r_·*t* = 1 − (1 − *x*)^1/3^(3)

Furthermore, when the solid–liquid reaction is controlled by both the diffusion reaction and interfacial chemical reaction, the leaching kinetics equation of zinc oxide dust particles may be expressed as follows:*k*_0_·*t* = 1/3ln(1 − *x*) − [1 − (1 − *x*)^−1/3^](4)
where *k*_d_ is the diffusion rate constant of the solid–liquid phase reaction, *k*_r_, the constant of solid–liquid interfacial chemical reaction; *k*_0_, the reaction rate constant under mixed solid–liquid control, *x*, the leaching rate of zinc from zinc oxide dust, and *t*, the leaching time.

To define the procedure for controlling an ultrasound-enhanced leaching process, a kinetic study was conducted for the ultrasound-enhanced leaching of zinc oxide dust. Relevant data from the experiment concerning the effect of a leaching time on the leaching rate of zinc from zinc-bearing metallurgical dust were put into Equations (2)–(4), respectively, for plotting the curves of 1 − 2/3*x* − (1 − *x*)^2/3^, 1 − (1 − *x*)^1/3^ and 1/3ln(1 − *x*) − 1+ (1 − *x*)^−1/3^ vs. time *t* (0–20 min), and the results are as shown in Figure 11a–c, representing the curves of zinc leaching processes under the control of solid product layer diffusion, control of the interfacial chemical reaction and mixed control, respectively. 

It should be mentioned that the experimental data for the initial stage of the process (0–2 min) are ignored in Figure 11a–c. The purpose was to reduce the data disturbance caused by the uncontrolled transfer process in the initial stage. Bringing relevant data into the model will make it difficult for the fitting results to correctly reflect the kinetic conditions of the main reaction process. Therefore, the 2–15 min stage is selected for fitting the experimental data. Similar research methods were applied in the study of Wang et al. [39] and Gui et al. [41].

The apparent reaction rate constants at different leaching temperatures obtained by the fitting (*k*_d_, *k*_r_, and *k*_0_) and the fitting coefficients related to the kinetic equations of reaction rates are as shown in Table 2.

In addition, the reaction rate constant k before the zinc leaching process approaching equilibrium (i.e., a fitting equation) at different temperatures was obtained according to Figure 11a–c and substituted into the Arrhemus empirical equation, respectively, as follows:*K* = *A*·exp(−*Ea*/*RT*)(5)
where *E**a* represents the activation energy of reaction, kJ/mol, *A* represents the frequency factor as a constant, *T* represents the temperature (*K*), and *R* represents the gas constant, 8.314 × 10^−3^ kJ/(mol·K).

Take the logarithm for both sides of Equation (4) to obtain the relation between ln*k* and 1/T:Ln*k* = ln*A* − *Ea*/*RT*(6)

A curve of ln*k* vs. 1/*T* is plotted as shown in Figure 12. The degree of fitting for the lnk vs. 1/*T* curve under mixed control (*R*^2^ = 0.9684) was significantly higher than that under diffusion control (*R*^2^ = 0.2315) and interfacial chemical reaction control (*R*^2^ = 0.3929), further indicating that the ultrasound-enhanced leaching of zinc oxide dust was primarily controlled by the mixed control of solid product layers, with an initial apparent activation energy of reaction 6.90 kJ/mol.

### 3.3. Comparative Experiment of Conventional-Ultrasonic Leaching

The optimal zinc leaching conditions for zinc oxide dust under an ultrasonic field determined under time conditions are as follows: leaching temperature 25 °C; ultrasonic power 500 W; sulfuric acid concentration 140 g/L; liquid–solid ratio 5:1; rotating speed 100 rpm; and leaching time 30 min, for which the zinc leaching rate can reach 91.16%. Under the optimal process conditions, a comparison between conventional stirring and ultrasound enhancement for an effect on the leaching rate of the zinc from oxide dust was conducted. The results are as shown in Figure 13: the zinc leaching rate under conventional stirring was 85.36%, and was increased by 5.8% under ultrasound enhancement at 500 W.

### 3.4. Analysis of Leaching Mechanism

#### 3.4.1. Characterization by XRD

The XRD patterns of the zinc oxide dust and leached residue are as shown in Figure 14. Obviously, the ZnO phase was mostly dissolved after sulfuric acid leaching, and for the conventional leached residue, there was still a ZnO peak. The reason for this might be that there were wraps in the zinc oxide dust, and the ZnO in the wrapped particles was not leached. For both ultrasonic leached residue and conventional leached residue, the ZnO peak intensity was weakened or the peak disappeared as compared with the raw material. Zinc in the residues existed in forms of ZnFe_2_O_4_, Zn_2_SiO_4_, and ZnS accompanied by large amounts of PbSO_4_, PbS, and SiO_2_.

#### 3.4.2. Comparative Analysis of SEM-EDS

For the raw material, the conventional leached residue and ultrasonic leached residue underwent SEM morphology characterization, for which the results are shown in Figure 15. Figure 15 shows that the zinc oxide dust particles were under a state of aggregation, and under the conventional leaching conditions, the wrapped particles had no change in morphology with a corrosion phenomenon on their surfaces. However, under the ultrasound enhancement, the leached residue had particles dispersed evenly, and the wrapped particles and the wraps were opened. This is also an important reason why the zinc leaching rate under ultrasound is higher than that under conventional leaching conditions. An analysis of leached residue by EDS spot scanning was carried out, and the results were as shown in Figure 15 and Figure 16. The leached residue was complicated in composition, with the coexistence of Zn, Pb, Fe, O, S, and Si.

#### 3.4.3. Particle Size Analysis and Mechanism of Leaching ZOD

Table 3 lists the detailed particle size parameters of the ZOD sample (A), the conventional leaching residue (B), and the ultrasonic leaching residue (C). As presented in Table 3, compared with the raw ZOD sample, the parameter values of D_10_, D_50_, D_90,_ and D_av_ are increased and the values of the surface area-to-volume ratio are decreased for the conventional leaching residue. The change of parameters may be due to the disappearance of the ZnO phase mainly distributed in the fine particles after leaching, while a large number of encapsulated particles remain in the conventional leaching residue. However, the parameter values of D_10_, D_50_, D_90,_ and D_av_ are decreased and the values of surface area-to-volume ratio are increased for the ultrasonic leaching residue compared with the raw ZOD sample. The change in parameters may be due to the generation of a large number of bubbles under the action of ultrasonic cavitation. With the growth and burst of the bubbles, huge energy is released, which promotes the inclusion of the ZnO phase surface to fall off, and realizes the dissociation of the encapsulated particles to generate a large number of tiny particles. The schematic diagram of the leaching mechanism of ultrasonic-enhanced ZOD particles is shown in Figure 17.

In summary, combined with the above analysis results of zinc leaching efficiency, XRD characterization, SEM-EDS characterization, and particle size analysis, it can be inferred that the leaching efficiency of zinc from zinc oxide dust (ZOD) was improved by ultrasonic strengthening treatment.

## 4. Conclusions

(1)Through an ultrasound-enhanced sulfuric acid leaching experiment, the optimal zinc leaching conditions for zinc oxide dust were determined as follows: leaching temperature of 25 °C; ultrasonic power of 500 W; sulfuric acid concentration of 140 g/L; liquid–solid ratio of 5:1; rotating speed of 100 rpm; and a leaching time of 30 min—for which the zinc leaching rate could reach up to 91.16%.(2)In a kinetic analysis of the ultrasound-enhanced sulfuric acid leaching of zinc oxide dust, the initial apparent activation energy of the reaction was 6.90 kJ/mol. indicating that the ultrasound-enhanced leaching of zinc oxide dust was primarily controlled by the mixed control of the solid product layers.(3)The leached residue was characterized by XRD and SEM-EDS, and the results showed that with ultrasonic waves, the encapsulated mineral particles were dissociated, and the dissolution of ZnO was enhanced. Mostly, the zinc in the leached residue existed in the forms of ZnFe_2_O_4_, Zn_2_SiO_4_, and ZnS accompanied by large amounts of PbSO_4_, PbS, and SiO_2_.

## Figures and Tables

**Figure 1 materials-15-05969-f001:**
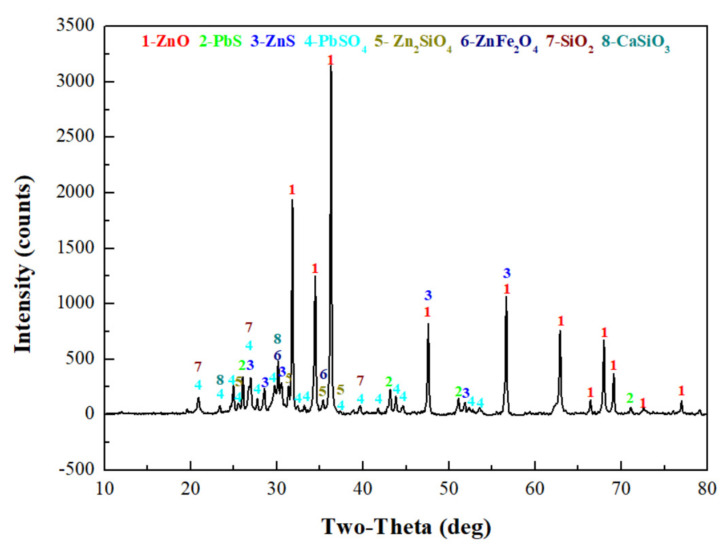
XRD pattern of the raw ZOD sample.

**Figure 2 materials-15-05969-f002:**
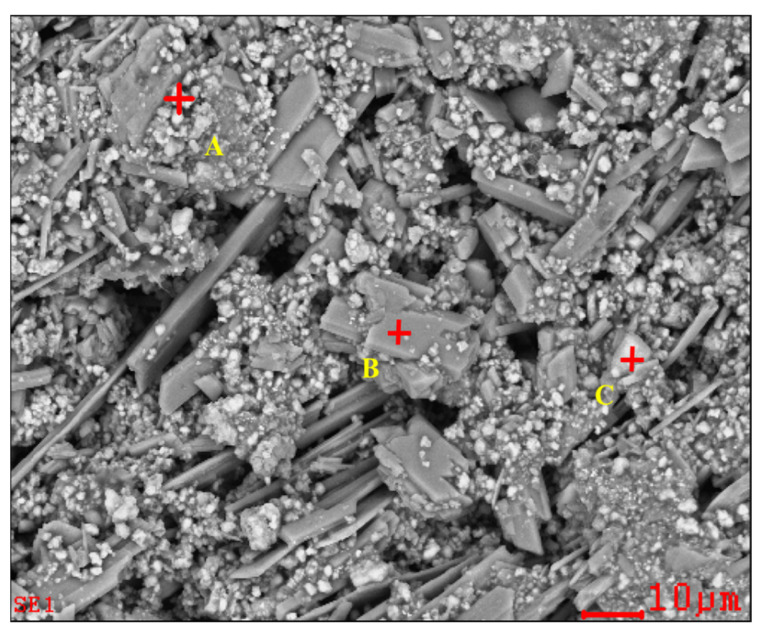
SEM image and EDS point analysis of the raw ZOD sample.

**Figure 3 materials-15-05969-f003:**
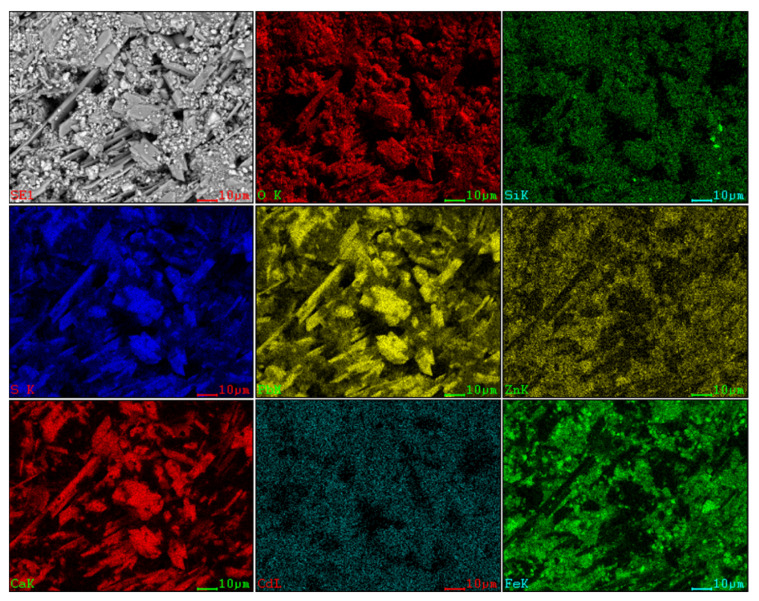
SEM image and EDS mapping analysis of the raw ZOD sample.

**Figure 4 materials-15-05969-f004:**
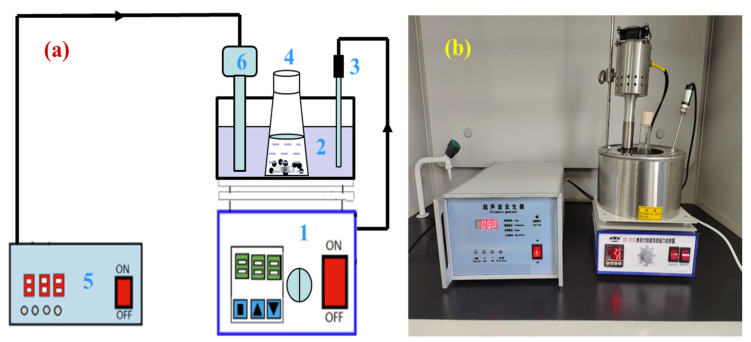
Experimental apparatus for ultrasound leaching: (**a**) schematic diagram; and (**b**) device diagram. In (**a**), 1—heat collection type thermostatic bath; 2—water bath; 3—thermometer, 4—conical flask; 5—ultrasonic generator; 6—ultrasonic probe.

**Figure 5 materials-15-05969-f005:**
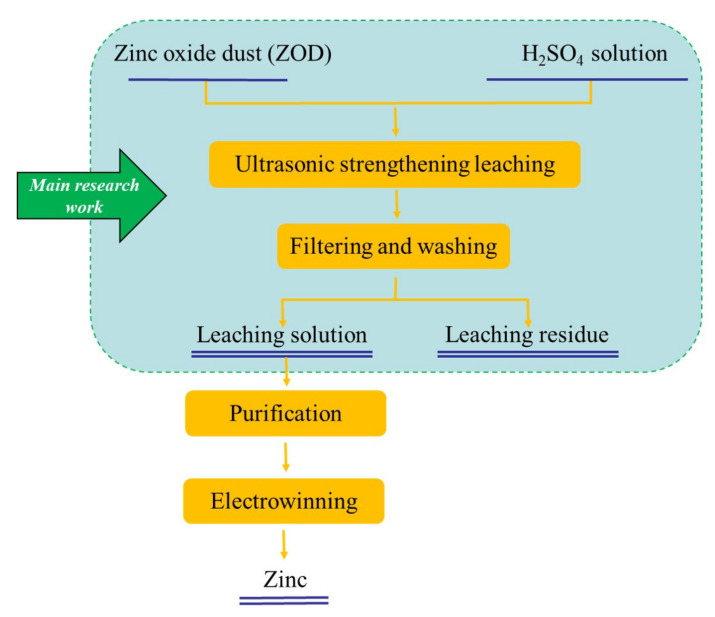
Flow diagram of the leaching process for ZOD.

**Figure 6 materials-15-05969-f006:**
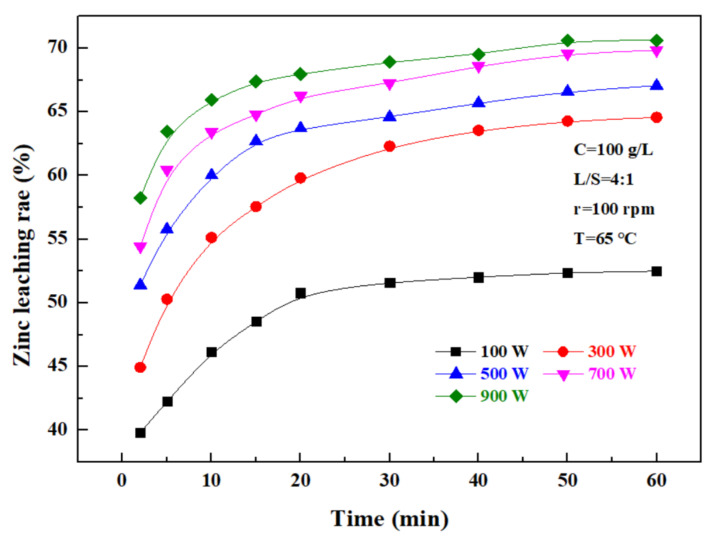
Effect of ultrasonic power on zinc leaching rate.

**Figure 7 materials-15-05969-f007:**
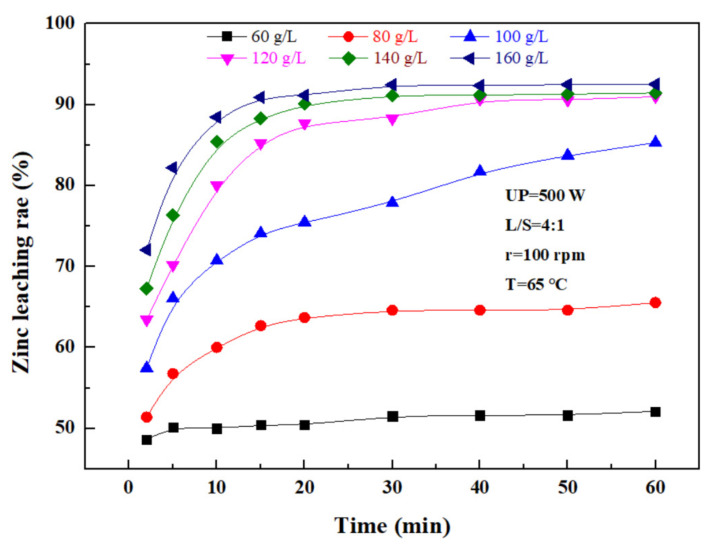
Effect of the sulfuric acid concentration on the zinc leaching rate.

**Figure 8 materials-15-05969-f008:**
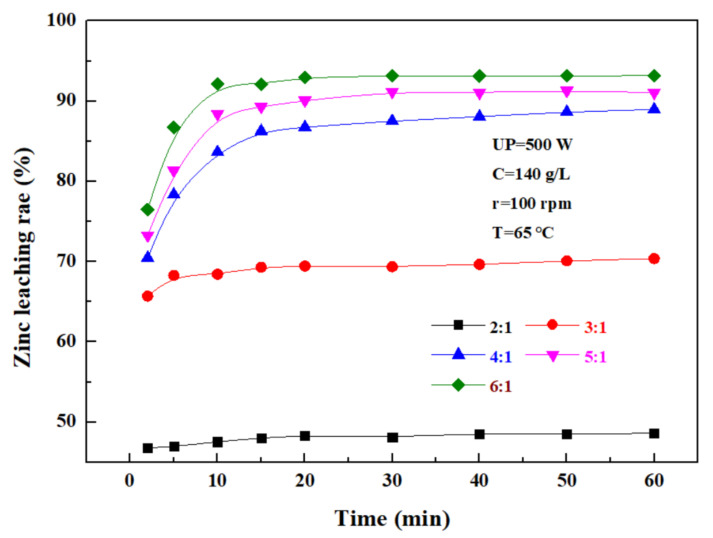
Effect of liquid–solid ratio on zinc leaching rate.

**Figure 9 materials-15-05969-f009:**
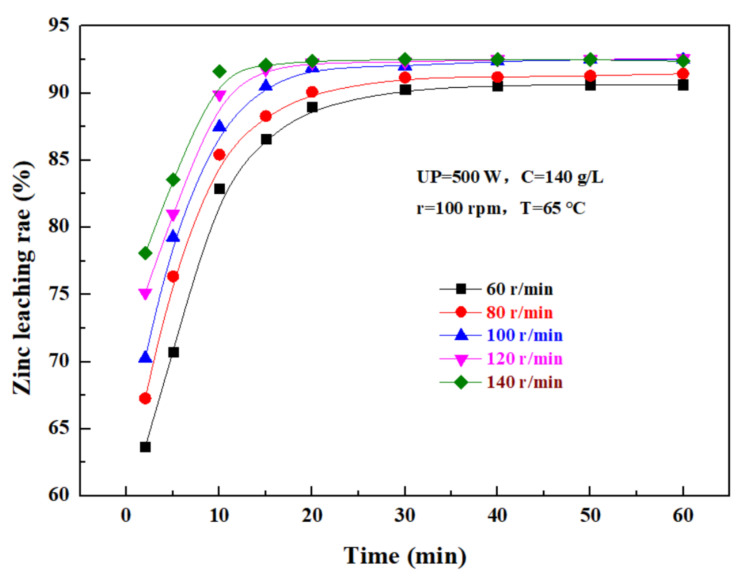
Effect of rotating speed on zinc leaching rate.

**Figure 10 materials-15-05969-f010:**
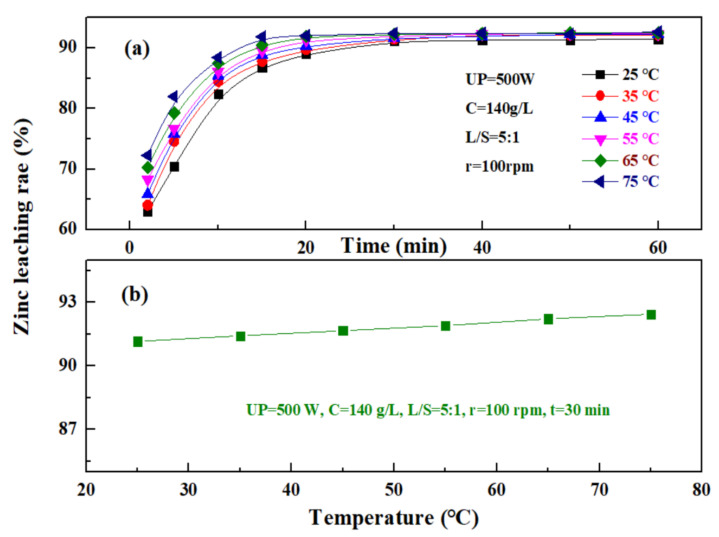
Effect of temperature on zinc leaching rate: (**a**)—different times; and (**b**)—30 min.

**Figure 11 materials-15-05969-f011:**
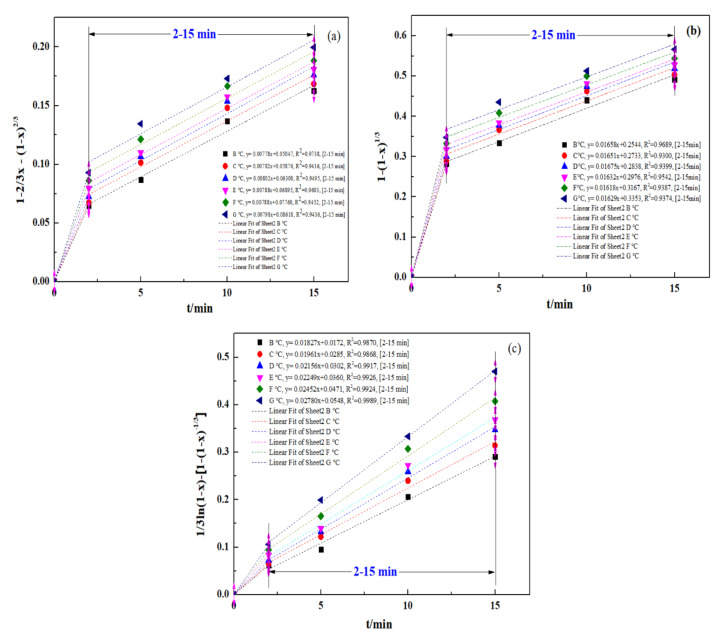
Plot of 1 − 2/3*x* − (1 − *x*) ^2/3^, 1 − (1 − *x*)^1/3^, and 1/3ln(1 − x)-[1 − (1 − x)^−1/3^] vs. time for various temperatures: (**a**) 1 − 2/3*x* − (1 − *x*) ^2/3^; (**b**) 1 − (1 − *x*)^1/3^; and (**c**) 1/3ln(1 − x) − [1 − (1 − *x*)^−1/3^].

**Figure 12 materials-15-05969-f012:**
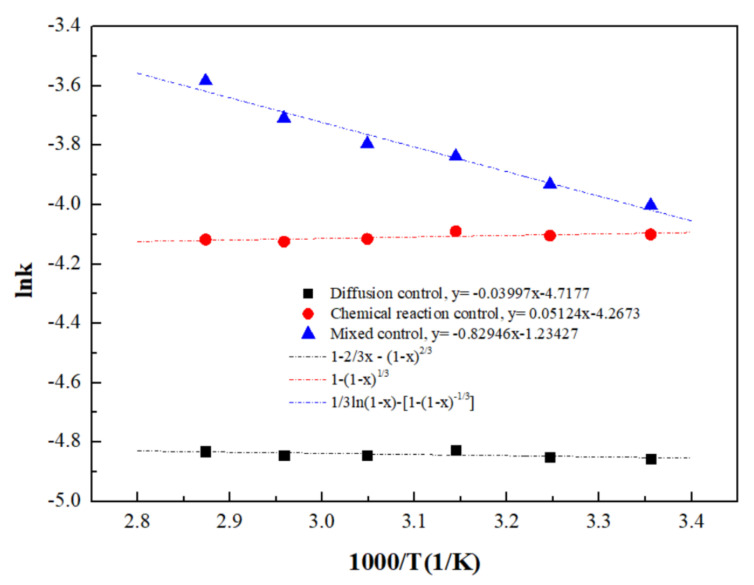
Arrhenius curve obtained for the dissolution of ZOD.

**Figure 13 materials-15-05969-f013:**
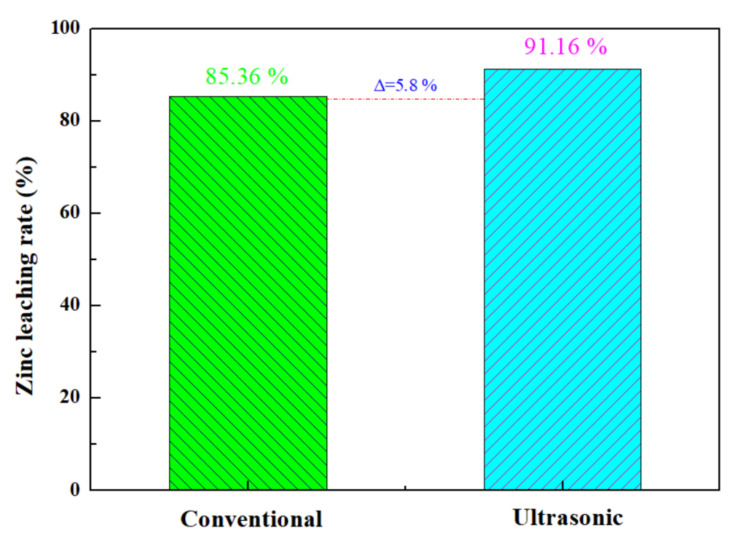
Comparison of zinc leaching rate between conventional and ultrasonic treatment of zinc oxide dust.

**Figure 14 materials-15-05969-f014:**
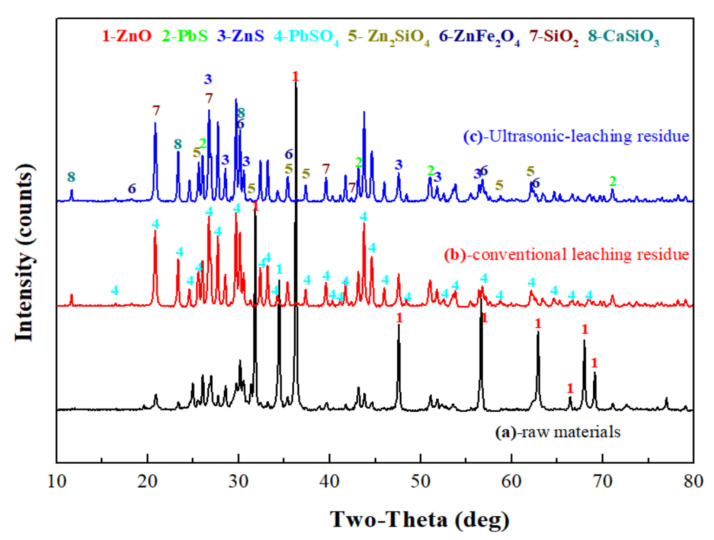
XRD patterns of raw materials (**a**), conventional leaching residue (**b**), and ultrasonic-leaching residue (**c**).

**Figure 15 materials-15-05969-f015:**
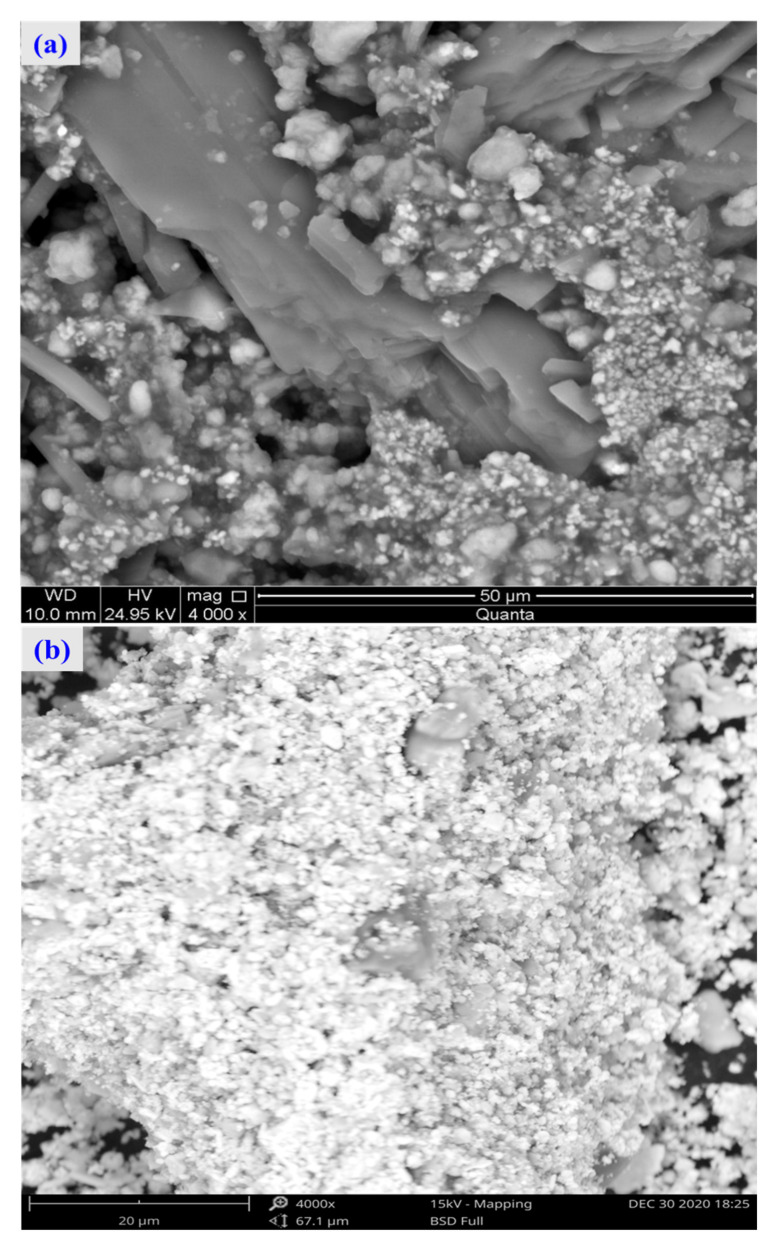
SEM patterns of raw materials (**a**), conventional leaching residue (**b**), and the ultrasonic leaching residue (**c**).

**Figure 16 materials-15-05969-f016:**
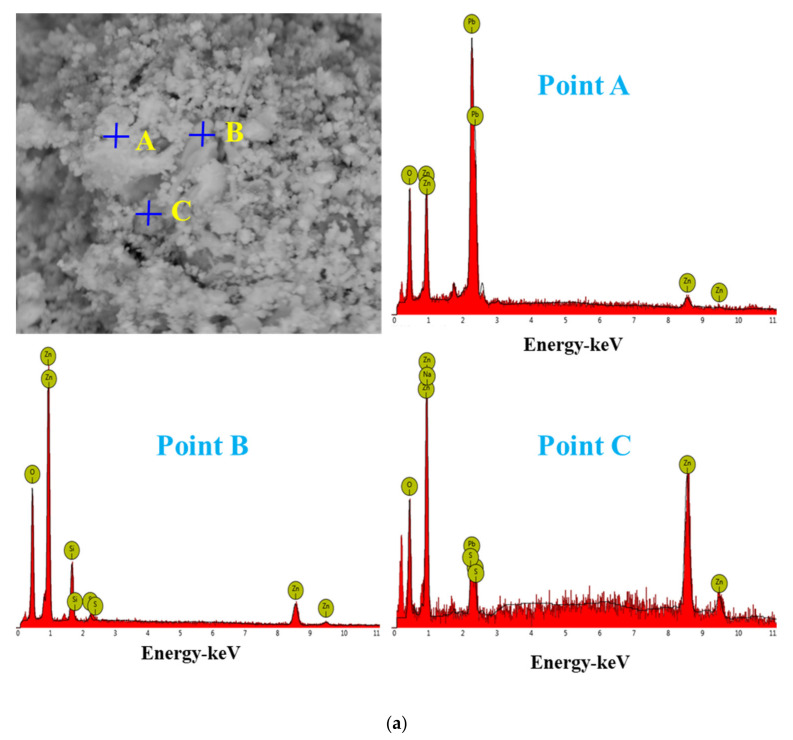
EDS point analysis of the conventional leaching residue (**a**), and the ultrasonic leaching residue (**b**).

**Figure 17 materials-15-05969-f017:**
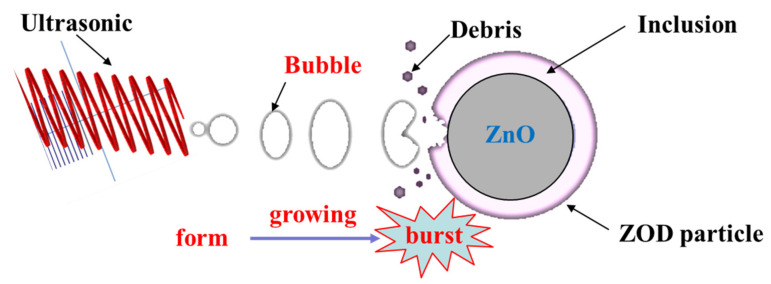
The schematic diagram of the leaching mechanism of ultrasonic-enhanced ZOD particle.

**Table 1 materials-15-05969-t001:** The chemical composition of the ZOD sample (mass fraction, %).

Element	Zn	Pb	Cd	Fe	Mn	S
**Content/%**	41.37	19.77	1.01	2.05	0.20	3.95
**Element**	Cl	Si	Ca	In	F	
**Content/%**	0.28	0.19	0.12	820.8 g/t	<0.01	

**Table 2 materials-15-05969-t002:** Correlation coefficients of fitting results for various models at various temperatures.

T (°C)	1 − 2/3*x* − (1 − *x*) ^2/3^		1 − (1 − *x*)^1/3^		1/3ln(1 − *x*) – 1 + (1 − *x*)^−1/3^	
	*k* _d_	*R* ^2^	*k* _r_	*R* ^2^	*k* _0_	*R* ^2^
**25**	0.00778	0.9738	0.01658	0.9689	0.01827	0.9870
**35**	0.00782	0.9416	0.01651	0.9300	0.01961	0.9868
**45**	0.00802	0.9495	0.01675	0.9399	0.02156	0.9917
**55**	0.00788	0.9603	0.01632	0.9542	0.02249	0.9926
**65**	0.00788	0.9452	0.01618	0.9387	0.02452	0.9924
**75**	0.00798	0.9436	0.01629	0.9374	0.02780	0.9989

**Table 3 materials-15-05969-t003:** Particle size parameters of the ZOD sample (A), the conventional leaching residue (B), and the ultrasonic leaching residue (C).

Samples	D_10_ (μm)	D_50_ (μm)	D_90_ (μm)	D_av_ (μm)	Surface Area-to-Volume Ratio (m^2^/cm^3^)
A	0.821	1.047	1.230	1.031	5.9886
B	1.038	1.355	1.608	1.336	4.6482
C	0.725	0.871	1.007	0.861	7.0916

## Data Availability

The data presented in this study are available upon request from the corresponding author. The data are not publicly available due to technical or time limitations.

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
