# Peer review of "A Study on the Mechanism and Kinetics of Ultrasound-Enhanced Sulfuric Acid Leaching for Zinc Extraction from Zinc Oxide Dust"

_materials, 2022, doi:10.3390/ma15175969_

Round 1

Reviewer 1 Report

The manuscript is interesting, but I consider that must be restructured with the aim of enhancing the quality of the approach and results.    

I recommend the following:

·       Review the language.

·   Reconsider the context in which the word “obviously” is used. I some statements is inadequate and in other undermines the idea being expressed.

·       Review the analysis and discussion of the results since there are contradictory approaches (see lines 164-170, section 3.1.2).

·       Further analysis is necessary. E.g., in section 3.1.5, the authors conclude that increasing the temperature results in the volatilization of the acid. Is it possible the prove this? Could it be that temperature has not a significant effect on the kinetics? 

Author Response

Dear Editor-in-Chief and reviewer:

Thank you for your letter and the reviewers’ comments concerning our manuscript entitled “ID materials-1853178 - A study on the mechanism and kinetics of ultrasound enhanced sulfuric acid leaching for zinc extraction from zinc oxide dust.” Those comments are all valuable and very helpful for revising and improving our paper, as well as the important guiding significance to our researches. We have studied comments carefully and have made correction which we hope meet with approval.

If this resubmission can meet the requirements of you and this journal now, please give us a chance and accept it. I would be greatly appreciated if you could spend some of your time on our resubmission. Looking forward to hearing from you soon.

Best regards!

The revised words, sentences, figures and tables highlighted in blue in this revised manuscript, and the statement of what has been done in this revised manuscript to each comment of the reviewers and the editor-in-chief is as follow:

Response to Reviewer 1 Comments:

Comment 1: Review the language.

Response: Thanks for the reviewer’s valuable and professional suggestion! According to the reviewer’s valuable and professional suggestion, we have changed the original manuscript language as shown in the revised manuscript, and the revised words, sentences, figures and tables highlighted in blue. Thanks for the reviewer’s valuable and professional suggestion again! Please check!

Comment 2: Reconsider the context in which the word “obviously” is used. I some statements is inadequate and in other undermines the idea being expressed.

Response: Thanks for the reviewer’s valuable and professional suggestion! According to the reviewer’s valuable and professional suggestion, we have revised the sentences in which the word “obviously” appears in 10 places in the text, as detailed in the revised manuscript.

Comment 3: Review the analysis and discussion of the results since there are contradictory approaches (see lines 164-170, section 3.1.2).

Response: Thanks for the reviewer’s valuable and professional suggestion! See lines 166, “… the zinc leaching rate was lowered because the hydrogen ion concentration in the reaction was increased with sulfuric acid concentration….”, the word “lowered” should be revised to “increased”. The revised places were marked in blue in section 3.1.2 in the revised version. Thanks for the reviewer’s valuable and professional suggestion again! Please check!

Comment 4: Further analysis is necessary. E.g., in section 3.1.5, the authors conclude that increasing the temperature results in the volatilization of the acid. Is it possible the prove this? Could it be that temperature has not a significant effect on the kinetics?

Response: Thanks for the reviewer’s valuable and professional suggestion! According to the reviewer’s valuable and professional suggestion, we have replotted and analyzed the Figure 10, as detailed in the revised manuscript.

Fig. 10(a) shows that the change of temperature had a significant effect on the zinc leaching rate when the time was 0 to 20 min, and the zinc leaching rate increased with the increase of temperature. At different temperatures, the zinc leaching rate increased rapidly with time at first, and reached a plateau gradually after 20-30 min. This is due to that with the increase of leaching temperature, the leaching reaction rate was advanced accordingly, and at the same time, the viscosity of solution decreased, which was conducive to the diffusion of leaching solvent and product, and thus, zinc leaching rate was significant at the beginning, while at a later stage, with the continuous consumption of leaching agent, the leaching efficiency was lowered gradually, and further increase of leaching temperature did not affect the dissolution of zinc oxide dust obviously. Fig. 10(b) shows that the zinc leaching rate of zinc oxide dust was less affected by temperature after the leaching time reaches 30 min, and the zinc leaching rates were 91.16% and 92.44% at 25 ℃ and 75 ℃, respectively. Considering that high temperatures may increase the volatilization of acid, resulting in a high acid consumption and increased economic cost, the leaching temperature was controlled at 25℃, and the 30min zinc leaching rate reached 91.16%.

The revised places were marked in blue in section 3.1.5 in the revised version. Thanks for the reviewer’s valuable and professional suggestion again! Please check!

Figure 10. Effect of temperatures on zinc leaching rate, (a)- different times; (b)-30 min.

Reviewer 2 Report

Manuscript ID: materials-1853178

Title: A study on the mechanism and kinetics of ultrasound enhanced sulfuric acid leaching for zinc extraction from zinc oxide dust

Authors: Xuemei Zheng et al.

Introduction.

Line 70-71. The Introduction section contains almost no description of the various hydrometallurgical methods for leaching of zinc oxide. It is necessary to add more information, show the extraction degree and the optimal parameters of each of the methods, and make a comparison of the results obtained.

Line 72-87. The authors described the positive aspects of ultrasonic enhanced. However, there are also negative ones – a high rate of erosion of the sonotrode, both from cavitation and from the influence of acid. Why not just use a high-pressure reactor to intensify the leaching process?

Materials and methods.

Section 2.3 must be re-named to “acid leaching tests” or something else.

Section 2.3. Add more information about ultrasonication system (scheme or photo with description). It is very important information, without it this article doesn't make sense.

Results

Section 3.1.1 How was changed the solution temperature during different ultrasound enhanced (100-900W). I think the temperature must be significant increased.

Figure 4-8. Authors must add the information about leaching process to figures title or image (T, concentration, L/S ratio, ultrasound effect, etc.)

Figure 9. A very common mistake was made in the calculations. In figure 4-8 there is no point 0, and in figure 9 the curves do not pass-through point 0, it simply is not there. This is a mistake; authors need to redo the calculation. Authors can read more information here: Levenspiel O. Chemical Reaction Engineering. https://the-seventh-dimension.com/images/textlev/LEVENSPIEL%20Chemical%20reaction%20engineering-ch1-ch2.pdf

Figure 12. Add information to figures title about a) b) and c).

Section 3.2 How was changes the particle size distribution of raw material and residue after two leaching methods (conventional and ultrasound).

This article about “mechanism and kinetics”, so authors must add chemical equation of leaching process. Authors must add separate section with discuss mechanism of leaching. It is necessary to add image with mechanism description.

The references are not formed by Materials style, please improve it.

Technical errors:

All chemical compounds and ions are spelled incorrectly. It is necessary to use a register for the valency and the number of atoms in the molecules of substances. Authors need to check the entire text of the article.

Author Response

Dear Editor-in-Chief and reviewer:

Thank you for your letter and the reviewers’ comments concerning our manuscript entitled “ID materials-1853178 - A study on the mechanism and kinetics of ultrasound enhanced sulfuric acid leaching for zinc extraction from zinc oxide dust.” Those comments are all valuable and very helpful for revising and improving our paper, as well as the important guiding significance to our researches. We have studied comments carefully and have made correction which we hope meet with approval.

If this resubmission can meet the requirements of you and this journal now, please give us a chance and accept it. I would be greatly appreciated if you could spend some of your time on our resubmission. Looking forward to hearing from you soon.

Best regards!

The revised words, sentences, figures and tables highlighted in blue in this revised manuscript, and the statement of what has been done in this revised manuscript to each comment of the reviewers and the editor-in-chief is as follow:

Response to Reviewer 2 Comments:

Comment 1: Line 70-71. The Introduction section contains almost no description of the various hydrometallurgical methods for leaching of zinc oxide. It is necessary to add more information, show the extraction degree and the optimal parameters of each of the methods, and make a comparison of the results obtained.

Response: Thanks for the reviewer’s valuable and professional suggestion! According to the reviewer’s valuable and professional suggestion, we have refined various hydrometallurgical methods for leaching zinc oxide, described in detail about the extraction process and optimal parameters for each method, and performed a comparative analysis of the results obtained. The revised places were marked in blue in section 1 Introduction in the revised version. Thanks for the reviewer’s valuable and professional suggestion again! Please check!

Comment 2: Line 72-87. The authors described the positive aspects of ultrasonic enhanced. However, there are also negative ones – a high rate of erosion of the sonotrode, both from cavitation and from the influence of acid. Why not just use a high-pressure reactor to intensify the leaching process?

Response: Thanks for the reviewer’s valuable and professional suggestion! We have characterized and analyzed the raw materials of zinc oxide fume, and also conducted conventional acid leaching experiments in the early stage. It was found that the zinc oxide fume produced by the volatilization furnace is a typical encapsulated material, and the zinc phase and the gangue component are encapsulated with each other, resulting in low leaching efficiency of conventional leaching. Ultrasound can strengthen the dissociation of inclusions, accelerate the chemical reaction and improve the zinc leaching rate, so as to achieve the purpose of efficient zinc extraction. However, the main function of pressure leaching is to speed up the reaction rate, and the effect of improving the zinc leaching rate of wrapped materials is slightly insufficient compared with ultrasonic.

Thanks for the reviewer’s valuable and professional suggestion again!

Comment 3: Section 2.3 must be re-named to “acid leaching tests” or something else.

Response: Thanks for the reviewer’s valuable and professional suggestion!  According to the reviewer’s valuable and professional suggestion, we have replaced the word “Procedure” by “Experimental methods”. The revised places were marked in blue in Section 2.3 in the revised revision. Thanks for the reviewer’s valuable and professional suggestion again! Please check!

Comment 4: Section 2.3. Add more information about ultrasonication system (scheme or photo with description). It is very important information, without it this article doesn't make sense.

Response: Thanks for the reviewer’s valuable and professional suggestion! According to the reviewer’s valuable and professional suggestion, We have supplemented the ultrasonic enhanced leaching experimental apparatus(Fig 4), process flow diagram(Fig 5) and made relevant instructions. The revised places were marked in blue in Section 2.3. Experimental methods in the revised revision. Thanks for the reviewer’s valuable and professional suggestion again! Please check!

Comment 5: Section 3.1.1 How was changed the solution temperature during different ultrasound enhanced (100-900W). I think the temperature must be significant increased.

Response: Thanks for the reviewer’s valuable and professional suggestion! During the ultrasonic leaching process, with the increase of ultrasonic power or the prolongation of ultrasonic leaching time, the temperature of the water bath will increase to a certain extent. In order to ensure that the leaching temperature remains unchanged, we adjusted the temperature of the water bath to ensure that the temperature deviation of the water bath was within 2 °C. The revised places were marked in blue in Section 2.3. Experimental methods in the revised revision. Thanks for the reviewer’s valuable and professional suggestion again! Please check!

Comment 6: Figure 4-8. Authors must add the information about leaching process to figures title or image (T, concentration, L/S ratio, ultrasound effect, etc.)

Response: Thanks for the reviewer’s valuable and professional suggestion! We have added the information about leaching process to figures title or image, as detailed in the revised Figure 6-10. Thanks for the reviewer’s valuable and professional suggestion again! Please check!

Comment 7: Figure 9. A very common mistake was made in the calculations. In figure 4-8 there is no point 0, and in figure 9 the curves do not pass-through point 0, it simply is not there. This is a mistake; authors need to redo the calculation. Authors can read more information here: Levenspiel O. Chemical Reaction Engineering. https://the-seventh-dimension.com/images/textlev/LEVENSPIEL%20Chemical%20reaction%20engineering-ch1-ch2.pdf

Response: Thanks for the reviewer’s valuable and professional suggestion! The unreacted shrinkage core model used in this paper is a classic model for leaching kinetics research, which is widely used in the leaching process of metals such as Zn, Fe, Cu, Au, etc., and the corresponding research papers have been published in recognized journals, such as Hydrometallurgy, Chemical Engineering Journal, Journal of Central South University, Separation Science and Technology, etc., and these research methods are consistent with the research methods of this paper. In addition, some of our previous work on kinetics research have also been recognized by experts in related fields and published in academic journals. The following lists the same kinetics research cases as this paper, and reviewers can view them through the link.

S.Aydogan. et al. Dissolution kinetics of sphalerite in acidic ferric chloride leaching, Chemical Engineering Journal.( https://doi.org/10.1016/j.cej.2005.09.005)

Qihao Gui. et al. The ultrasound leaching kinetics of gold in the thiosulfate leaching process catalysed by cobalt ammonia, Hydrometallurgy. (https://doi.org/10.1016/j.hydromet.2020.105426)

Zhi Xiong Liu. et al. Leaching and kinetic modeling of calcareous bornite in ammonia ammonium sulfate solution with sodium persulfate, Hydrometallurgy. (http://dx.doi.org/10.1016/j.hydromet.2014.01.011)

WANG Rui-xiang, et al. Leaching kinetics of low grade zinc oxide ore in NH3-NH4Cl-H2O system, Journal of Central South University.( https://doi.org/10.1007/s11771-008-0126-4)

Ma Aiyuan, et al. Clean recycling of zinc from blast furnace dust with ammonium acetate as complexing agents, Separation Science and Technology, (https://doi.org/10.1080/01496395.2018.1444057)

Thanks for the reviewer’s valuable and professional suggestion again!

Comment 8: Figure 12. Add information to figures title about a) b) and c).

Response: Thanks for the reviewer’s valuable and professional suggestion! We have added the information about a) b) and c), as detailed in the revised Figure 14. Thanks for the reviewer’s valuable and professional suggestion again! Please check!

Comment 9: Section 3.2 How was changes the particle size distribution of raw material and residue after two leaching methods (conventional and ultrasound).

Response: Thanks for the reviewer’s valuable and professional suggestion! We have added the information about particle size parameters of the ZOD sample, the conventional leaching residue and the ultrasonic leaching residue, as detailed in the revised Section 3.4.3. Particle size analysis and mechanism of leaching ZOD. Thanks for the reviewer’s valuable and professional suggestion again! Please check!

Comment 10: This article about “mechanism and kinetics”, so authors must add chemical equation of leaching process. Authors must add separate section with discuss mechanism of leaching. It is necessary to add image with mechanism description.

Response: Thanks for the reviewer’s valuable and professional suggestion! We have added the information about the schematic diagram of the leaching mechanism of ultrasonic-enhanced ZOD particle, as detailed Figure 17 in Section 3.4.3. Particle size analysis and mechanism of leaching ZOD. In addition, in this well-known research field, the leaching reaction equation of ZnO in sulfuric acid (ZnO + H2SO4 = ZnSO4 + H2O) is not necessary. Because the study did not involve any other reactions. Thanks for the reviewer’s valuable and professional suggestion again! Please check!

Comment 11: The references are not formed by Materials style, please improve it.

Response: Thanks for the reviewer’s valuable and professional suggestion! The format of some references is not standardized, and we have revised them according to the requirements of the journal format. The revised places were marked in red in Section References in the revised revision. Thanks for the reviewer’s valuable and professional suggestion again! Please check!

Comment 12: All chemical compounds and ions are spelled incorrectly. It is necessary to use a register for the valency and the number of atoms in the molecules of substances. Authors need to check the entire text of the article.

Response: Thanks for the reviewer’s valuable and professional suggestion! According to the reviewer’s valuable and professional suggestion, we have corrected all chemical compounds and ions. Some of the problems may be caused by the layout of the manuscript. The revised places were marked in blue in the revised revision. Thanks for the reviewer’s valuable and professional suggestion again! Please check!

Round 2

Reviewer 1 Report

The suggestions have been taken into account.

Author Response

Thanks for the reviewer’s valuable and professional suggestion! According to the reviewer’s valuable and professional suggestion, we have changed the original manuscript language as shown in the revised manuscript, and the revised words and sentences highlighted in red. The author has supplemented and refined the relevant references in the introduction section. The revised places were marked in red in Section 1. Introduction in the revised revision. At the same time, we have supplemented the diagram of the device for ultrasonic enhanced leaching (Fig 4),. The revised places were marked in red in Section 2.3. Experimental methods in the revised revision. Thanks for the reviewer’s valuable and professional suggestion again! Please check!

Reviewer 2 Report

The authors answered many questions, but there are a number of comments:

1) What is the shape of the sonotrode? This is an important point, it is not clear from figure 4. Add a photo of experimental equipment if it is possible.

2) I will repeat my remark about the incorrect calculation of kinetics. Authors should not cite articles with errors. When calculating the kinetics, authors must use the point "0". I'm sure the Guest Editor of the special issue – Prof. Dr. Guo Chen will agree with my arguments.

Author Response

Response to Reviewer 2 Comments:

 Comment 1: What is the shape of the sonotrode? This is an important point, it is not clear from figure 4. Add a photo of experimental equipment if it is possible.

Response: Thanks for the reviewer’s valuable and professional suggestion! According to the reviewer’s valuable and professional suggestion, we have supplemented the diagram of the device for ultrasonic enhanced leaching (Fig 4),. The revised places were marked in red in Section 2.3. Experimental methods in the revised revision. Thanks for the reviewer’s valuable and professional suggestion again! Please check!

Comment 2: I will repeat my remark about the incorrect calculation of kinetics. Authors should not cite articles with errors. When calculating the kinetics, authors must use the point "0". I'm sure the Guest Editor of the special issue – Prof. Dr. Guo Chen will agree with my arguments.

Response: Thanks for the reviewer’s valuable and professional suggestion! According to the reviewer’s valuable and professional suggestion, we performed recalculations and graphs, as well as illustrating the initial stages of the reaction. The revised places were marked in red (lines 289-301) in Section 3.2. Kinetics of ultrasound enhanced leaching in the revised revision. Thanks for the reviewer’s valuable and professional suggestion again! Please check!